# A Nodes-Based Non-Structural Model Considering a Series Structure for Heat Exchanger Network Synthesis

**Yue Xu, Heri Ambonisye Kayange and Guomin Cui \***

School of Energy and Power Engineering, University of Shanghai for Science and Technology, 516 Jungong Road, Shanghai 200093, China; yue323@foxmail.com (Y.X.); ambonisye@duce.ac.tz (H.A.K.)
\* Correspondence: cgm@usst.edu.cn

**Abstract:** The aim of heat exchanger network synthesis is to design a cost-effective network configuration with the maximum energy recovery. Therefore, a nodes-based non-structural model considering a series structure (NNM) is proposed. The proposed model utilizes a simple principle based on setting the nodes on streams such that to achieve optimization of a heat exchanger network synthesis (HENS) problem. The proposed model uses several nodes to quantify the possible positions of heat exchangers so that the matching between hot and cold streams is random and free. Besides the stream splits, heat exchangers with series structures are introduced in the proposed model. The heuristic algorithm used to solve NNM model is a random walk algorithm with compulsive evolution. The proposed model is used to solve four scale cases of a HENS problem, the results show that the costs obtained by NNM model can be respectively lower 3226 \$/a(Case 1), 11,056 \$/a(Case 2), 2463 \$/a(Case 3), 527 \$/a(Case 4) than the best costs listed in literature.

**Keywords:** NNM model; stream splits; series matches; HENS design; random walk with compulsive evolution

---

## 1. Introduction

In chemical engineering, energy of the process system engineering is mainly consumed by heating or cooling devices. Considering a comprehensive design problem of system integration is essential to maximum energy recovery, such as in ethanol reforming area [1], in residential buildings [2], and in heat exchanger network synthesis [3].

Heat exchanger network synthesis is an important part of processing systems, and searching the optimal solution could save the cost and energy consumption. Along with the development of computational skills, mathematical programming has been a dominant tool for solving the increasing complex optimization problems that include plenty of continuous and integer variables. Additionally, mathematical programming has proved its advantages in the design of heat exchanger network synthesis [4–6]. Moreover, in the design of heat exchanger network, a simplified and reliable Synheat model can help to search the desire solution [7]. Among the optimized models, the stage-wise superstructure (SWS) model of Yee and Grossmann [8] is a widely used foundational model for solving the heat exchanger network synthesis (HENS) problems. Based on the SWS model, many retrofitted SWS models have been proposed in recent years [9–16]. The common characteristic of these structural models is that the generation of heat exchangers happens in a certain fixed stage, which could provide a distinct description of HENS configurations. These models have certain limitations on the quality and efficiency of HENS design. Firstly, structural models exclude some of the potential structures because of their fixed structures [17], thus restricting the searching ability of heuristic algorithms. Secondly, these structural models can impose difficulty in optimization computation, especially in large-scale HENS design, a number of potential structures are spiked, which can introduce difficulty

in the algorithm. In order to extend the solution domain, the main retrofitted methods based on structural models make the model structure more complex or add stages to the network, but these methods can achieve satisfactory results. However, the computational efficiency decreases with the expansion of the stages of cases. To overcome the mentioned bottleneck of the HENS design, a novel optimized model that can guarantee quality and efficiency is necessary. Therefore, we proposed a nodes-based non-structural model considering a series structure(NNM) model that uses nodes on streams to quantify the position of heat exchangers in the network. The generation mode of the proposed model is relatively free due to the randomly matching nodes on hot and cold streams without structural limit. Because of the characteristic of NNM model above, only adding some nodes on streams could increase many potential solutions, but in SWS model, adding optimization stages is necessary. Thus, NNM model reduces the computational burden while achieving reliable results.

After the series structures on a substream branch added to the HENS design were presented by Jongsuwat et al. [18], many researchers have implemented them in their models and proved their necessity in HENS design. Floudas et al. [19,20] used the serial matches on a substream in the transshipment model of Papoulias and Grossmann [21]. Huang [22] combined the advantages of a hyperstructure and SWS model proposing a new model wherein multiple utilities were placed at each stage. In addition, heat exchangers were placed in series, and stream bypass was also embedded in this model. Kim [23] proposed a stages/substages superstructure model based on the SWS model, introducing the concept of substages into the SWS model and serial heat exchangers, as well as stream bypass. Based on [23], Pavão et al. [15] proposed a new superstructure model containing sub-splits, stream bypass, serial heat exchanger, and multi utilities. Galli and Cerdá [24] considered heat exchangers placed in series to tackle the HENS problem with stream splits. The existing literature concerning series heat exchangers on a substream is still based on structural models, and optimization difficult in structural models still exist in these models.

In this article, NNM which includes nodes quantification, stream splits, and series structure is proposed. The proposed model can extend the solution domain's freedom while yielding a candidate structure. Furthermore, we use the Random walk algorithm with compulsive evolution (RWCE) [25], a robust heuristic algorithm to optimize the proposed model because the RWCE has proved its practicability in searching for an optimal solution when being applied to the non-structural model [26]. The performances of the proposed model are evaluated using four different cases, and the obtained results are compared to the results presented in the literature to prove the validity of the proposed model.

## 2. Model and methods

### 2.1. Problem Statement

Obtaining the minimal total annual cost (TAC) represents the objective function in a heat exchanger network synthesis problem. In order to address this, some known conditions should be given, such as a set of hot and cold streams with their temperatures, heat capacities ($MCp$), and convective heat transfer coefficients ($h$). Besides, because of using heat exchangers to achieve the heat transfer between hot and cold streams, the utilities such as heaters and coolers are also needed for calculating the TAC. Moreover, the area costs of heat exchangers and the unit cost of utilities are also necessary. In the existing literature, the valves, fitting, and piping costs of stream splits are negligible.

### 2.2. Nodes-based Non-structural Model

Direct model optimization determines the optimization quality. Obtaining better solutions under the premise of not increasing the model complexity is important. Accordingly, we propose a new NNM model with stream splits and series structure. The main advantages of our model are as follows. Firstly, NNM model has more flexible setting pattern, series structure is allowed in NNM model, besides, the number of series nodes can be freely set without any limitation. Secondly, the stage

concept is discarded, and the positions of heat exchangers are quantified by the positions of nodes, thus, more flexible matching pattern than other model, so that it could offer many potential structures for later optimization. Thirdly, only variables concerning nodes are used for positions' qualifying, which improves the computational efficiency. Fourthly, in real life engineering, the series or parallel heat exchanger structures always exist in a system. NNM model could search the best solution in a shorter computational time, which is beneficial for saving cost and resources, so it fits better real-life cases.

In this paper, some definitions about NNM model should be introduced first. The number of stream branches is named as $Nf_H$, $nf_H = 1, 2, \ldots, Nf_H$ on each hot stream while $Nf_C$, $nf_C = 1, 2, \ldots, Nf_C$ on each cold stream. $Nf_H$ and $Nf_C$ can be set according to the scale of cases in NNM model. Because the series structure is allowed in the NNM model, some nodes are located on each stream branch. The number of nodes on each hot stream branch is denoted as $Mb_H$, $mb_H = 2, 3, \ldots, Mb_H$, while the number of nodes on each cold stream branch is denoted as $Mb_C$, $mb_C = 2, 3, \ldots, Mb_C$. Thus, the number of nodes in a stream splits group is equal to $Mb_H \times Nf_H$ on hot stream, and $Mb_C \times Nf_C$ on cold stream. Besides, several stream-splits groups are set on each stream for maintaining multi times matching. Setting $Nd_H$, $nd_H = 1, 2, \ldots, Nd_H$ as the number of aforementioned stream-splits groups on each hot stream, and $Nd_C$, $nd_C = 1, 2, \ldots, Nd_C$ as the number of stream-splits groups on each cold stream. Furthermore, $Ne_{N_H}$ and $Ne_{N_C}$ denote the numbers of nodes on each hot stream and each cold stream, respectively, and they are respectively calculated by Equations (1) and (2).

$$Ne_{N_H} = Mb_H \times Nf_H \times Nd_H, ne_i = 2, 3, \ldots, Ne_{N_H}, i \in N_H \tag{1}$$

$$Ne_{N_C} = Mb_C \times Nf_C \times Nd_C, ne_j = 2, 3, \ldots, Ne_{N_C}, j \in N_C \tag{2}$$

$Nt_H$ and $Nt_C$ are used to express the total number of nodes on all the hot and cold streams, respectively, and they are respectively calculated by the following formulas. The suffixes *H* and *C* respectively relate to hot and cold, which identify the property of variables.

$$Nt_H = Ne_{N_H} \times N_H \tag{3}$$

$$Nt_C = Ne_{N_C} \times N_C \tag{4}$$

The schematic representation of NNM model with stream splits is shown in Figure 1, wherein it can be seen that it takes a small-scale case containing two hot streams and two cold streams. As Figure 1 shows, nodes are placed on each stream branch, utilities are placed at the end of streams, and only one kind of utility is applied to each stream.

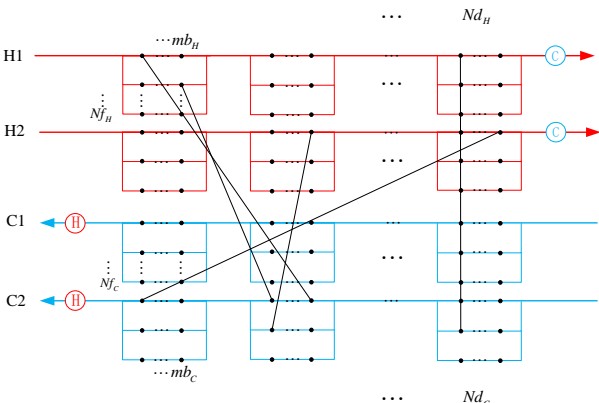

**Figure 1.** The schematic representation of NNM with series structure.

When a heat exchanger is generated in a network, it can be expressed by matching a pair of nodes that have not been matched before on hot stream and cold stream, respectively. Therefore, the position

of the generated heat exchanger can be expressed by the serial number node. For later calculation, the serial number on the hot stream is denoted as $Mn_H$ and on the cold stream as $Mn_C$. The relationship between $Mn_H$, $Nd_H$, $nf_H$, and $Mb_H$ is given by Equation (5), and the relationship between $Mn_C$, $nd_C$, $nf_C$, and $Mb_C$ is given by Equation (6).

$$Mn_H = (i-1) \times Ne_{N_H} + (nd_H - 1) \times (Mb_H \times Nf_H) + (nf_H - 1) \times Mb_H + mbn_H$$
$$mbn_H \in [1, Mb_H], \ Mn_H \in Nt_H \tag{5}$$

$$Mn_C = (j-1) \times Ne_{N_C} + (nd_C - 1) \times (Mb_C \times Nf_C) + (nf_C - 1) \times Mb_C + mbn_C$$
$$mbn_C \in [1, Mb_C], \ Mn_C \in Nt_C \tag{6}$$

The $mbn_H$ is denoted as the serial number of nodes on each hot stream branch, while $mbn_C$ is denoted as the serial number of nodes on each cold stream branch. Due to the mapping relation between $Mn_H$ and $Mn_C$, these two variables can be transferred to each other, as given by Equations (7) and (8).

$$NL_{Mn_H} = Mn_C \tag{7}$$

$$NL_{Mn_C} = Mn_H \tag{8}$$

For two heat exchangers located on the same stream branch, the split ratio of the corresponding branch is related to the specific stream, and position of stream-splits group and stream branch. Hence, the split ratios of hot and cold branches in each stream-splits group, which are denoted as *SPH* and *SPC*, are expressed by Equations (9) and (10), respectively.

$$\sum_{nf_H=1}^{Nf_H} SPH_{i,nd_H,nf_H} = 1.0, \ i \in N_H, \ nd_H = 1, 2, \ldots, Nd_H \tag{9}$$

$$\sum_{nf_C=1}^{Nf_C} SPC_{j,nd_C,nf_C} = 1.0, \ j \in N_C, \ nd_C = 1, 2, \ldots, Nd_C \tag{10}$$

### 2.3. Model-Related Calculation

In this sub-section, the calculation method that concerns the inlet and outlet temperatures and the split ratio of each branch is introduced. The calculation of temperature starts from the inlet temperature of a hot stream. The inlet temperature of a parent stream at the number of stream-splits groups of $nd_H$ equals to the outlet temperature at the number of stream-splits groups of $(nd_H - 1)$. Especially when $nd_H$ is equal to 1, the corresponding inlet temperature equals to the stream inlet temperature. Regarding the temperatures on stream splits, the inlet temperature of the first node on each stream branch is the same and equal to the one on the parent stream. Due to the series structure, the inlet temperature of all the nodes on a stream branch except for the first one equals the outlet temperature of their previous node. These relationships between the temperatures are given by Equations (11)–(13).

$$T^{out}_{H,i,nd_H} = T^{in}_{H,i,(nd_H+1)} \tag{11}$$

$$T^{in}_{H,i,nd_H} = T^{in}_{H,i,nd_H,nf_H,1}, \ nf_H = 1, 2, \ldots, Nf_H \tag{12}$$

$$T^{in}_{H,i,nd_H,nf_H,mb_H} = T^{out}_{H,i,nd_H,nf_H,(mb_H+1)} \tag{13}$$

In addition, the calculation of the inlet and outlet temperatures of a node and the calculation of the outlet temperature of the parent stream and the ones on stream branches are shown as following formulas (Equations (14) and (15)).

$$T^{out}_{H,i,nd_H,nf_H,mb_H} = T^{in}_{H,i,nd_H,nf_H,mb_H} - Q_{Mn_H} / (MCp_i \times SPH_{i,nd_H,nf_H}), mb_H = 1, 2, \ldots, Mb_H, i \in N_H \tag{14}$$

$$T^{out}_{H,i,nd_H} = \sum_{nf_H=1}^{Nf_H} (T^{out}_{H,i,nd_H,nf_H,Mb_H} \times SPH_{i,nd_H,nf_H}), i \in N_H \tag{15}$$

Considering the situation on a cold stream, the related formulas of the temperatures on a cold stream are given by Equations (16)–(20).

$$T^{out}_{C,j,nd_C} = T^{in}_{C,j,(nd_C-1)} \tag{16}$$

$$T^{in}_{C,j,nd_C} = T^{in}_{C,j,nd_C,nf_C,Mb_C}, nf_C = 1,2,\ldots,Nf_C \tag{17}$$

$$T^{in}_{C,j,nd_C,nf_C,mb_C} = T^{out}_{C,j,nd_C,nf_C,(mb_C+1)} \tag{18}$$

$$T^{out}_{C,j,nd_C,nf_C,mb_C} = T^{in}_{C,j,nd_C,nf_C,mb_C} + Q_{NLC(Mn_C)}/(MCp_j \times SPC_{j,nd_C,nf_C}), mb_C = 1,2,\ldots,Mb_C, j \in N_C \tag{19}$$

$$T^{out}_{C,j,nd_C} = \sum_{nf_C=1}^{Nf_C} (T^{out}_{C,j,nd_C,nf_C,1} \times SPC_{j,nd_C,nf_C}), j \in N_C \tag{20}$$

As can be noticed, Equations (16)–(20) are similar to Equations (11)–(15), but there are order differences that should be paid attention to in the calculation process. For instance, the calculation of node temperature begins from the inlet temperature of a cold stream, so the calculation goes from the later node to the former node on each stream branch. As Equation (17) presents, the inlet temperature of a later node on each stream branch is the same as the inlet temperature of the parent stream.

*2.4. Objective Function*

The objective function is defined by Equation (21).

$$\text{TAC} = \sum_{i=1}^{N_H} (F_{fix} + C_A \cdot A^{\beta}_{HU,i}) \cdot Z_{HU,i} + \sum_{j=1}^{N_C} (F_{fix} + C_A \cdot A^{\beta}_{CU,j}) \cdot Z_{CU,j} + $$
$$\sum_{Mn_H=1}^{Nt_H} (F_{fix} + C_A \cdot A^{\beta}_{Mn_H}) \cdot Z_{Mn_H} + \sum_{i=1}^{N_H} C_{CU} \cdot Q_{CU,i} + \sum_{j=1}^{N_C} C_{HU} \cdot Q_{HU,j} \tag{21}$$

The objective function is composited of the cost of heat exchangers and utilities. Among these costs, $F_{fix}$ represents the fixed capital cost, $A$ is the area of a heat exchanger, $C$ and $\beta$ denote the area coefficient and its exponent, respectively; $Z$ denotes a binary variable representing the existence of costs, and $Q$ stands for the heat load. The suffixes *HU* and *CU* relate to the short for hot utility and cold utility, respectively; suffixes $i$ and $j$ denote the specific hot stream and cold stream, respectively; lastly, suffix $Mn_H$ relates to the serial number of a hot node. In addition, $N_H$ and $N_C$ are the numbers of hot streams and cold streams, respectively. The formula for calculating the area of heat exchangers is given by Equation (22), where $U_{i,j}$ denotes the overall heat transfer efficiency.

$$A_{Mn_H} = \frac{Q_{Mn_H}}{U_{i,j} \cdot LMTD_{i,j}}, i \in N_H, j \in N_C \tag{22}$$

The temperature difference used in this paper is Logarithmic mean temperature difference (*LMTD*). The supplementary formulas of $U_{i,j}$ and $LMTD_{i,j}$ are given by Equations (23)–(27).

$$U_{i,j} = h_i \cdot h_j / (h_i + h_j), \ i \in N_H, j \in N_C \tag{23}$$

$$\theta_1 = T^{in}_{H,Mn_H} - T^{out}_{C,Mn_C} \tag{24}$$

$$\theta_2 = T^{out}_{H,Mn_H} - T^{in}_{C,Mn_C} \tag{25}$$

$$LMTD_{Mn_H} = \frac{\theta_1 - \theta_2}{\ln(\theta_1/\theta_2)} \tag{26}$$

$$AMTD_{Mn_H} = \frac{1}{2}(\theta_1 + \theta_2) \tag{27}$$

Especially when $\theta_1$ equals to $\theta_2$, the *LMTD* is substituted by Arithmetic mean temperature difference (*AMTD*), and the specific expression is given by Equation (27). The suffixes $Mn_H$ and $Mn_C$ of the temperature in Equations (24) and (25) denote the serial numbers of hot node and cold node, respectively, which are matched to one heat exchanger.

*2.5. Constraints*

This sub-section introduces the constraints of temperatures, which satisfies the heat flow in the network. The constraints of coolers and heaters are defined by Equations (28) and (29), respectively.

$$(T_{H,i}^{out} - T_{H,i}^{target})MCp_i = Q_{CU,i}, \; i \in N_H \tag{28}$$

$$(T_{C,j}^{target} - T_{C,j}^{out})MCp_j = Q_{HU,j}, \; j \in N_C \tag{29}$$

In Equations (28) and (29), $T_{H,i}^{out}$ and $T_{C,j}^{out}$ represent the outlet temperatures of heat exchangers, and if there is a difference from the target temperature of a stream, the cooler or heater is used to satisfy the heat balance of the stream. The constraints about temperatures of each heat exchanger are given by Equations (30) and (31). If Equations (30) and (31) are not satisfied, then the matching is not suitable for the network. In this paper, this problem is solved by using a pair of utilities to substitute an infeasible heat exchanger.

$$T_{H,Mn_H}^{in} - T_{C,Mn_C}^{out} \geq 0, \; Mn_H \in Nt_H \tag{30}$$

$$T_{H,Mn_H}^{out} - T_{C,Mn_C}^{in} \geq 0, \; Mn_C \in Nt_C \tag{31}$$

*2.6. Methodology*

The RWCE can simultaneously handle integer and continuous variables by altering the heat load of heat exchangers. The individuals participating in the optimization are isolated, which results in high population diversity. Besides, the RWCE can postpone premature convergence when searching for an optimal solution to a HENS problem. The RWCE was introduced in detail in [27]. The RWCE flowchart is presented in Figure 2.

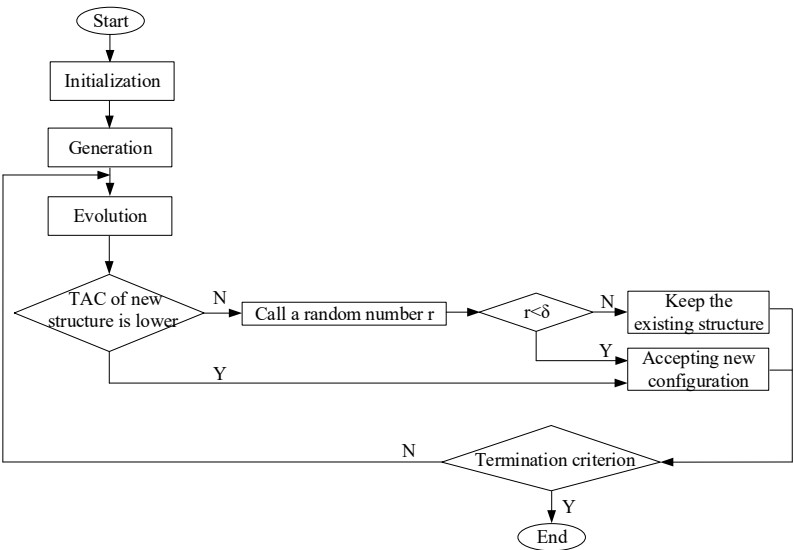

**Figure 2.** The random walk algorithm with compulsive evolution (RWCE) flowchart.

## 3. Proposed Model Validation

In this part, NNM model with stream splits having two nodes on each stream branch and the RWCE is evaluated through four cases from literature to verify the optimization ability of the proposed model. The first two cases containing the heat exchangers in series are used to evaluate the model's ability to search better solution. All the four cases are of different scales, and they have a lower cost than the other cases in the existing literature. The results of all the cases show that the proposed model is a powerful and robust tool with universal usage in solving the heat exchanger network synthesis problems. The code was implemented using Compaq Visual Fortran Version 6.6, and run on a Windows Server system with an Intel(R) Xeon(R) Gold 6140 CPU at 2.29-2.30 GHz.

### 3.1. Case Study 1 (4SP)

Case 1 was a small-scale case containing two hot and two cold streams. It was introduced by Linnhoff et al. [27]. The data of case 1 are shown in Table 1, and the comparison results of this case are shown in Table 2.

**Table 1.** Stream, cost, and parameters of case 1.

| Stream | Tin (K) | Tout (K) | $MCp$ (kW/K) | $h$ (kW/m$^2$/k) |
|---|---|---|---|---|
| H1 | 443 | 333 | 30 | 1.6 |
| H2 | 423 | 303 | 15 | 1.6 |
| C1 | 293 | 408 | 20 | 1.6 |
| C2 | 353 | 413 | 40 | 1.6 |
| HU | 450 | 450 | | 4.8 |
| CU | 293 | 313 | | 1.6 |

Annual cost of heat exchanger = $1000 \times A^{0.6}$\$/a($A$ in m$^2$)
Annual cost of hot utility = 80 \$/kW/a
Annual cost of cold utility = 20 \$/kW/a

**Table 2.** Comparison results of case 1.

| Reference | TAC (\$/a) | Units | $Q_H$ (kW) | $Q_C$ (kW) |
|---|---|---|---|---|
| Yee and Grossmann [8] | 80,274 | 5 | 0 | 400 |
| This work (Figure 3) | 77,958 | 5 | 0 | 400 |
| This work (Figure 4) | 77,048 | 5 | 0 | 400 |

Yee and Grossmann [8] optimized this case by the SWS model, and the result of 80,274 \$/a was achieved. Analyzing the solution to this case given in [8], four heat exchangers with two stream branches were applied to the network, and only one cold utility was located on the second hot stream. To evaluate NNM model ability, we first used the RWCE and the SWS model with stream splits to optimize this small case. The corresponding structure is shown in Figure 3. The result was 77,958 \$/a. Comparing the structure with the one in [8], a totally different solution was obtained, which contained only one stream branch. Besides the structure, the result obtained by NNM model applying the RWCE is presented in Figure 4. The nodes were set as $nd_H = nd_C = 2$, $Mb_H = Mb_C = 2$, $Nd_H = Nd_C = 2$. The TAC in Figure 4 is 77,048 \$/a, which is lower 910 \$/a than that in Figure 3, it was achieved in 521.92 s. It is interesting that the number of heat exchangers, the number of stream branches, the split ratio value, the cold-utility heat load, and its position were the same, while the heat load of heat exchangers was slightly different, as shown in Figures 3 and 4. The crucial reason for such results is that the two heat exchangers on the dotted line were on the same stream branch.

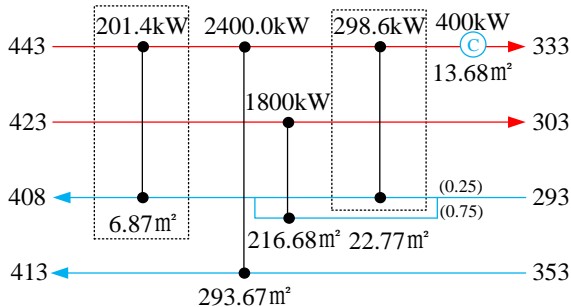

**Figure 3.** Solution of case 1 obtained by the stage-wise superstructure (SWS) model; the total annual cost (TAC) is 77,958 $/a.

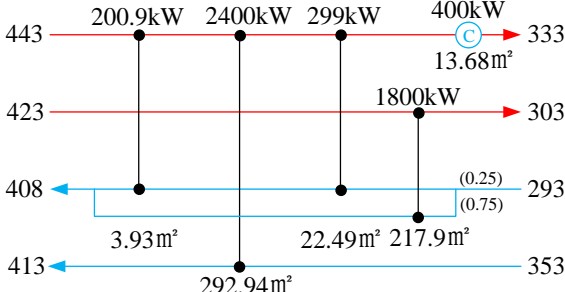

**Figure 4.** The structure obtained by NNM model with serial structures; the TAC is 77,048 $/a.

The piping cost of stream branches is not considered in a HENS problem with stream splits, so the final structures, having too many stream branches, are still not accepted in real engineering even when that decreases the system cost. Therefore, introducing the heat exchangers in series into NNM model could solve this kind of problem to a certain extent by adjusting the difference in temperatures of heat exchangers to obtain lower investment cost without increasing the number of stream branches.

### 3.2. Case Study 2 (6SP)

Case 2 was the case presented in the paper of Björk and Westerlund [11]. In this case, the numbers of hot and cold streams differed greatly; namely, there were two hot streams and four cold streams. Besides, this case represents a typical case which is suitable for a model with stream splits. The data of case 2 are shown in Table 3.

**Table 3.** Stream, cost, and parameters of case 2.

| Stream | $T_{in}$ (°C) | $T_{out}$ (°C) | $MCp$ (kW/(°C)) | $h$ (kW/m²/(°C)) |
|--------|---------------|----------------|-----------------|-------------------|
| H1 | 180 | 75 | 30 | 2 |
| H2 | 240 | 60 | 40 | 2 |
| C1 | 40 | 230 | 20 | 1.5 |
| C2 | 120 | 260 | 15 | 1.5 |
| C3 | 40 | 130 | 25 | 2 |
| C4 | 80 | 190 | 20 | 2 |
| HU | 325 | 325 | | 1 |
| CU | 25 | 25 | | 2 |

Annual cost of heat exchanger = $8000 + 50 \times A^{0.75}$ $/a(A in m²)
Annual cost of hot utility = 120 $/kW/a
Annual cost of cold utility = 6 $/kW/a

In the existing literature, the best solution to this case of 123,357 $/a was obtained by Huang et al. [28]. The framework used in their work was based on the multistage superstructure proposed by Huang et al. [22]. It needed six heat exchangers and two utilities along with three stream splits,

as well as a bypassing stream to accomplish the exchanging heat. Notably, the heater was placed on a stream branch of the second hot stream. This configuration totally exhibited the feature of their model. Bao and Cui [29] used the RWCE combined with the optimum-protection strategy to optimize this case and presented a solution in their paper. The model used in [29] denoted the modified SWS model permitting non-isothermal mixing, and Huang [30] obtained the TAC of 6SP is 128,169$/a. Comparing the structures and results provided in [28,29], it could be concluded that the heat exchangers in series were vitally important for enhancing the heat recovery and decreasing investment cost.

However, when NNM model was used to optimize this case, the solution shown in Figure 5 was obtained. As can be seen in Figure 5, only five heat exchangers and two utilities were used to exchange the heat. Moreover, only one stream branch was required. The structure shown in Figure 5 embodies the advantages of structures given in [19,28]. From the aspect of forbidding the stream bypass, the TAC of the structure presented in Figure 5 was 112,301 $/a, which was lower 11,056 $/a by that in [28]; and, the result was achieved in 7799.625 s. From the aspect of heat exchangers in series, the TAC of the structure presented in Figure 5 was 13,120 $/a, which was lower than the result presented in [29] that was optimized by the SWS and RWCE. Therefore, using heat exchangers in series could decrease the number of stream splits, which could be beneficial to the realization in practice. The comparison results are given in Table 4.

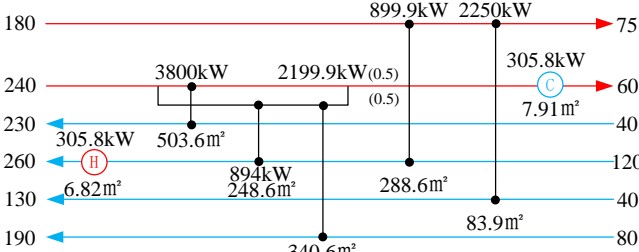

**Figure 5.** Solution of case 2 obtained by NNM model with serial structures; the TAC is 112,301 $/a.

**Table 4.** Comparison results of case 2.

| Reference | TAC ($/a) | Number of Units | $Q_H$ (kW) | $Q_C$ (kW) |
|---|---|---|---|---|
| Björk and Westerlund [5] | 139,083 | - | - | - |
| Bergamini et al. [19] | 140,367 | 10 | 314.5 | 315.0 |
| Bao and Cui et al. [29] | 125,421 | 8 | 304.5 | 304.5 |
| Huang et al. [30] | 128,169 | 8 | 315.0 | 315.0 |
| Huang et al. [28] | 123,357 | 8 | 315.0 | 315.0 |
| This work (Figure 5) | 112,301 | 7 | 305.8 | 305.8 |

### 3.3. Case Study 3 (15SP)

Case 3 is a large-scale case, having eight hot streams and seven cold streams. It was first proposed in the paper of Björk and Pettersson [31]. The data of 15SP are shown in Table 5.

Björk and his co-workers [5] solved this large-scale case by using a model based on the Synheat model [32], which was proposed in 2003. A hybrid algorithm based on the Genetic algorithm (GA) and deterministic Mixed Integer Nonlinear Programming approach (MINLP-approach) was used to solve this large-scale case. Peng and Cui [33] introduced a two-level procedure to optimize the candidate structure obtained by the SWS model without stream splits. The TAC of the possible structures was calculated on a lower level, and then these structures were evaluated by the Simulated annealing (SA). Fieg [34] formulated a monogenetic algorithm combined with the hybrid GA to seek an optimal design. Pavão et al. [35] optimized this case by combining the SWS model with a mixture model having the SA and Novel Rocket Fireworks Optimization (NRFO), while they obtained the TAC of 15SP by SA and particle swarm optimization (PSO) [36]. Then, they continuously enhanced an optimized method that

included the TAC and environmental impact [7]. Besides, they proposed a multi-objective model for solving a HENS problem.

**Table 5.** Stream, cost, and parameters of case 3.

| Stream | $T_{in}$ (°C) | $T_{out}$ (°C) | $MCp$ (kW/(°C)) | $h$ (kW/m²/(°C)) |
|--------|-----------|------------|-------------|----------------|
| H1 | 180 | 75 | 30 | 2 |
| H2 | 280 | 120 | 60 | 1 |
| H3 | 180 | 75 | 30 | 2 |
| H4 | 140 | 40 | 30 | 1 |
| H5 | 220 | 120 | 50 | 1 |
| H6 | 180 | 55 | 35 | 2 |
| H7 | 200 | 60 | 30 | 0.4 |
| H8 | 120 | 40 | 100 | 0.5 |
| C1 | 40 | 230 | 20 | 1 |
| C2 | 100 | 22 | 60 | 1 |
| C3 | 40 | 190 | 35 | 2 |
| C4 | 50 | 190 | 30 | 2 |
| C5 | 50 | 250 | 60 | 2 |
| C6 | 90 | 190 | 50 | 1 |
| C7 | 160 | 250 | 60 | 3 |
| HU | 325 | 325 | | 1 |
| CU | 25 | 40 | | 2 |

Annual cost of heat exchanger = $8000 + 50 \times A^{0.75}$ \$/a(A in m²)
Annual cost of hot utility = 80 \$/kW/a
Annual cost of cold utility = 10 \$/kW/a

The solution to this case obtained by NNM model is presented in Figure 6. Compared to the solution presented in [7], the numbers of heat exchangers and stream splits were the same. The difference was that the number of heaters in the solution presented in Figure. 6 was lower than that in [7]. Since the fixed capital cost of this case was 8000 \$/a, decreasing a heating unit made the TAC dramatically decreased. That is, the difficulty of eliminating heat exchangers existed in a later optimized period. Hence, the candidate structure obtained by NNM model during the optimization not only offered potential configurations to seek an optimal design but avoided optimal local design. The solution obtained in this work was 1,494,862 \$/a, and it was achieved in 8565.92 s, which denoted the lowest reported TAC for this case. The comparison results of this case are given in Table 6.

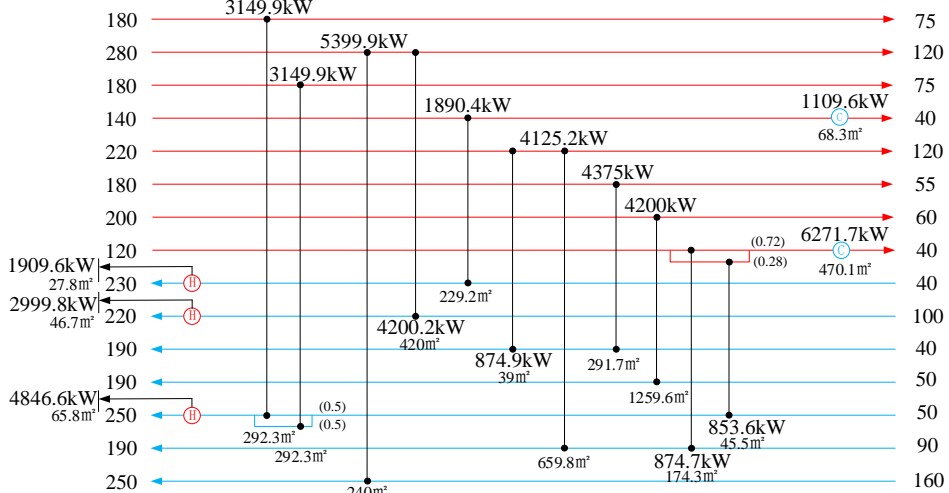

**Figure 6.** Solution of case 3 obtained by NNM with serial structures; the TAC is 1,494,862 \$/a.

**Table 6.** Comparison results of case 3.

| Reference | TAC ($/a) | Number of Units | $Q_H$ (MW) | $Q_C$ (MW) |
|---|---|---|---|---|
| Björk and Pettersson [32] | 1,513,854 | - | - | - |
| Björk and Nordman [5] | 1,530,063 | - | - | - |
| Peng and Cui [33] | 1,527,240 | 19 | 10.11 | 7.73 |
| Pavão et al. [36] | 1,525,394 | 19 | | |
| Fieg et al. [34] | 1,510,891 | 15 | 10.61 | 8.24 |
| Pavão et al. [35] | 1,507,290 | 19 | - | - |
| Pavão et al. [7] | 1,497,325 | 17 | - | - |
| This work (Figure 6) | 1,494,862 | 16 | 9.76 | 7.38 |

### 3.4. Case Study 4 (20SP)

This case was first presented by Xiao et al. [37]. The data of 20SP are shown in Table 7.

**Table 7.** Stream, cost, and parameters of case 4.

| Stream | $T_{in}$ (K) | $T_{out}$ (K) | $MCp$ (kW/K) | $h$ (kW/m²/k) |
|---|---|---|---|---|
| H1 | 453 | 348 | 30 | 2 |
| H2 | 553 | 393 | 15 | 0.6 |
| H3 | 453 | 348 | 30 | 0.3 |
| H4 | 413 | 318 | 30 | 2 |
| H5 | 493 | 393 | 25 | 0.08 |
| H6 | 453 | 328 | 10 | 0.02 |
| H7 | 443 | 318 | 30 | 2 |
| H8 | 453 | 323 | 30 | 1.5 |
| H9 | 553 | 363 | 15 | 1 |
| H10 | 453 | 333 | 30 | 2 |
| C1 | 313 | 503 | 20 | 1.5 |
| C2 | 393 | 533 | 35 | 2 |
| C3 | 313 | 463 | 35 | 1.5 |
| C4 | 323 | 463 | 30 | 2 |
| C5 | 323 | 523 | 20 | 2 |
| C6 | 313 | 423 | 10 | 0.06 |
| C7 | 313 | 423 | 20 | 0.4 |
| C8 | 393 | 483 | 35 | 1.5 |
| C9 | 313 | 403 | 35 | 1 |
| C10 | 333 | 393 | 30 | 0.7 |
| HU | 598 | 598 | | 1 |
| CU | 298 | 313 | | 2 |
| Annual cost of heat exchanger = 8000 + 50 × A$^{0.75}$ $/a(A in m²) | | | | |
| Annual cost of hot utility = 70 $/kW/a | | | | |
| Annual cost of cold utility = 10 $/kW/a | | | | |

Notably, it was needed to achieve the heat transfer between 10 hot streams and 10 cold streams, which was a challenge for mathematical model and algorithm optimization. Therefore, only several researchers studied this 20SP case. Luo [38] used a hybrid GA (genetic algorithm) and an explicit solution of stream temperatures with a stage-wise superstructure to solve this case. The design by Luo [38] and the one presented in Figure 7 has similar structures. The difference is that, in the design presented in Figure 7, the number of coolers is smaller, while the number of heat exchangers is larger compared to that in [36]. Laukkanena [39] presented a bi-level decomposition method to solve the multi-objective problem. Pavão et al. [35] presented a hybrid method mixing the SA and the NRFO to address this large-scale case. The mathematical model used in [34] was the SWS model, and three stages

were set to get $2.02 \times 10^9$ candidate structures. Among all the possible solutions, the TAC of the best one was 1,725,295 $/a.

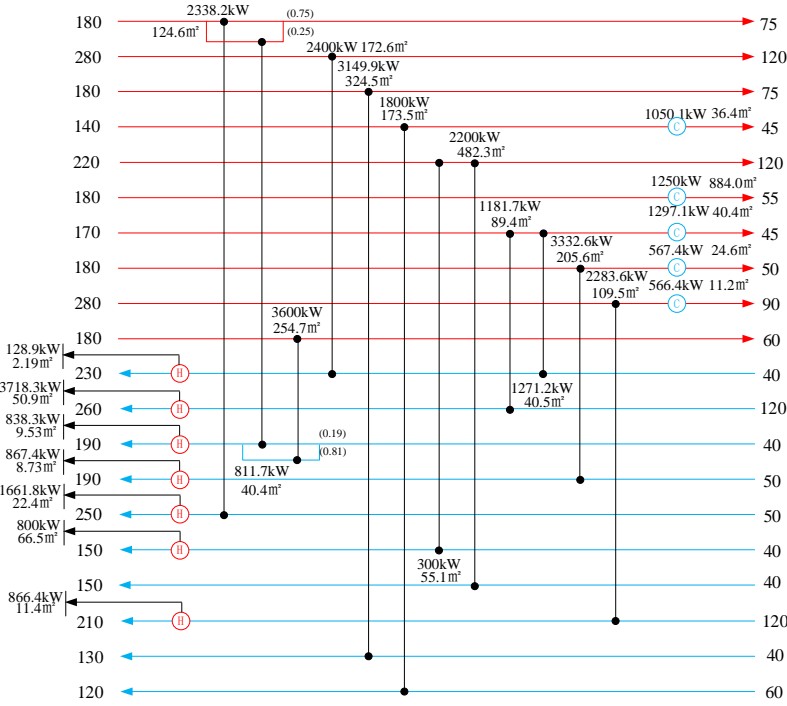

**Figure 7.** Solution of case 4 obtained by NNM model with serial structures; the TAC is 1,724,768 $/a.

Aiming to optimize this kind of large-scale cases, the optimization efficiency is as important as the results. In this large case, we set $Nf_H = Nf_C = 3$ and $Nd_H = Nd_C = 6$. Thus, only 32 nodes of each stream were enough to yield a better result. However, it might need three or four stages for obtaining a good result using the SWS model. Compared with the SWS model, the proposed model used much fewer variables in the evolution, which not only enhance the evolution efficiency but also decreased the structure cost. The comparison results are shown in Table 8, where it can be seen that the result obtained by NNM model is 1,724,768 $/a, and it was obtained in 8326.625 s. Consequently, we believe that the proposed NNM model can satisfy the requirements for both the results and efficiency of different scale cases.

**Table 8.** Comparison results of case 4.

| Reference | TAC ($/a) | Number of Units | $Q_H$ (MW) | $Q_C$ (MW) |
|---|---|---|---|---|
| Wu et al. [37] | 1,827,772 | 29 | 9.01 | 4.87 |
| Luo et al. [38] | 1,753,271 | 26 | 9.51 | 5.36 |
| Laukkanena and Tveita [39] | 1,739,778 | 24 | 9.50 | 5.35 |
| Pavão et al. [35] | 1,725,295 | 24 | - | - |
| This work (Figure 7) | 1,724,768 | 24 | 8.88 | 4.73 |

## 4. Conclusions

With the aim to develop a simple optimization model for solving complex HENS problems, a new NNM model with stream splits having heat exchangers in series is introduced in this paper. The generation of heat exchangers is quantified by randomly matching between hot and cold nodes. Hence, all possible matchings between hot and cold streams are used in this single-stage model. In addition, in this stream splits model, the heat exchangers in series are allowed. According to

the obtained results, the proposed model has many advantages compared to the existing models. The results of 4SP, 6SP, 15SP, and 20SP are lower by 3226 \$/a, 11,056 \$/a, 2463 \$/a, and 527 \$/a than the best-reported ones, respectively. The results of Case 1 (4SP) and Case 2 (6SP) showed that the proposed model not only decreased the investment cost under the premise that operation cost was unchanged but also achieved maximum heat load. In Case 3 (15SP) and Case 4 (20SP), which were without the heat exchangers in series, the proposed model provided more potential structures in the optimization process. Notably, the simplified structures obtained by the proposed model were similar to the ones obtained by the SWS model, which clearly indicated that the structures of the proposed model involved the ones of the SWS model, but had better configurations than those of the SWS model.

The cases used for the proposed model verification are of different scales, which implies that the proposed model can suit different-scale problems. However, there are still some limitations; namely, regardless a node is matched or not, the temperature calculation will include all the nodes in the network, which can increase the computational burden, but still, the proposed NNM model has higher computational speed compared with the SWS model. In order to obtain even better results, the proposed model can be further improved. For instance, the model can be realized such that when a node is not matched, it will be assumed that it is not a part of a network during the optimization process. Furthermore, the bypass flow and multi-utilities can be integrated into NNM model, which will be considered in our future work.

**Author Contributions:** Conceptualization, G.C.; software H.A.K.; provided the simulation case, writing Y.X.; All authors have read and agreed to the published version of the manuscript.

**Funding:** This research was funded by National Natural Science Foundation of China grand number 51176125, National Natural Science Foundation of China grand number 21978171 and Capacity Building Plan for some Non-military Universities and Colleges of Shanghai Scientific Committee grand number 16060502600.

**Conflicts of Interest:** The authors declare no conflict of interest.

## Nomenclature

*Abbreviation*

| | |
|---|---|
| AMTD | Arithmetic mean temperature difference |
| GA | Genetic algorithm |
| HENS | Heat exchanger network synthesis |
| LMTD | Logarithmic mean temperature difference |
| NNM | nodes-based non-structural model considering a series structure |
| NRFO | Novel Rocket Fireworks Optimization |
| MINLP | Mixed Integer Nonlinear Programming |
| SA | simulated annealing algorithm |
| SWS | stage-wise superstructure |
| TAC | total annual cost |
| RWCE | Random walk algorithm with compulsive evolution |

*Parameters of NNM*

| | |
|---|---|
| $A$ | Area, m$^2$ |
| $C_A$ | The coefficient of area cost, \$/m$^2$/a |
| $CU$ | Cold utility |
| $F_{fix}$ | The fixed capital cost, \$/a |
| $HU$ | Hot utility |
| $h$ | convective heat transfer coefficients |
| $MCp$ | heat capacities |
| $Mb_H$ | The number of nodes on each hot stream branch |
| $Mb_C$ | The number of nodes on each cold stream branch |
| $Mn_H$ | The serial number of hot node |
| $Mn_C$ | The serial number of cold node |
| $N_H$ | The number of hot streams |

| | |
|---|---|
| $N_C$ | The number of cold streams |
| $Ne_{N_H}$ | The number of nodes on each hot stream |
| $Ne_{N_C}$ | The number of nodes on each cold stream |
| $Nd_H$ | The number of stream-splits groups on each hot stream |
| $Nd_C$ | The number of stream-splits groups on each cold stream |
| $Nf_H$ | The number of stream branches on hot stream |
| $Nf_C$ | The number of stream branches on cold stream |
| $Nt_H$ | The total number of nodes on all hot streams |
| $Nt_C$ | The total number of nodes on all cold streams |
| $NL$ | The matching relationship between hot and cold nodes |
| $Q$ | Heat load, kW |
| $SPH$ | The split ratio on hot streams |
| $SPC$ | The split ratio on cold streams |
| $T$ | Temperature, °C |
| $U_{i,j}$ | overall heat transfer efficiency |
| $Z$ | The binary variable representing the existence of costs |
| $\beta$ | The exponent of area cost |
| $r$ | Random number |
| $\delta$ | The probability of accepting imperfect solution |

***Index***

| | |
|---|---|
| *in* | Inlet |
| *out* | Outlet |
| *target* | Target |
| *i* | Index of hot streams |
| *j* | Index of cold streams |

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
