# Peer review of "A Nodes-Based Non-Structural Model Considering a Series Structure for Heat Exchanger Network Synthesis"

_processes, doi:10.3390/pr8060695_

Round 1

Reviewer 1 Report

The paper presents development of optimization model for solving complex heat exchanger network synthesis problems. A new nodes-based non-structural model with hot and cold streams splits having heat exchangers in series was proposed. The model was tested and verified applying four different-scale problems. The content of the paper is very interesting and within the scope of the Processes journal. However, the paper needs major revision, i.e., especially thorough text editing should be done before acceptance.

Some (not all) problems to fix:

  • Line 12, 33, 34, 67, 70, 252, 253 and more – abbreviation HENS, SWS, NNM-SS, RWCE, GA, MINLP-approach, etc. should be introduced before first use.
  • Line 17 – “proposed model are used” – spelling mistake.
  • Line 26, 32 – capital letter in the middle of sentence.
  • 1 is illegible. It is hard to understand what is presented there.
  • Line 108, 109: References to eq. (1) – (4) were used before these equations were introduced. The same problem is also for all other equations. Please, introduce equation first and then refer to them.
  • (1) – (4) were not properly put into the text like they should be in the scientific paper. They were not introduced, and they stand away of the text. The same problem is for all other equations. General, equations should be introduced into the text in a smooth way.
  • Font size in Fig. 2 is too small.
  • Line 189: “[29]” should be not written as superscript. Similarly in line 192 and in the other places in the paper.
  • Line 192: “Yee and Grossmann optimized…” – reference number is needed after the names.
  • Line 200, 201: “…which is lower than that in Figure 3 that has the value of 910 $/a…” – unclear sentence.
  • Line 248: “having eight hot streams and seven hot streams” – unclear.
  • Line 256: „…SA (Simulated annealing)…” – please reformulate in the following way “…simulated annealing (SA)..”. The rule is that at first use the full name should be written then abbreviation should be introduced in the bracket. After that you can use abbreviation through entire paper. Similarly in line 258 for NRFO.
  • Line 261 “…nodes-based non-structural…” – you should use abbreviation NNM all the time instead of full name. The full name should be used only one time when the abbreviation is introduced. Similar problems were found in many places in the paper.
  • Line 277: “Notably, it was needed to achieve the heat transfer between 20 hot streams and 20 cold streams” – it is not clear. In table 7 10 hot and 10 cold streams are shown.

Reviewer 2 Report

1.The abstract is short and clear, it would be better if some main result of this research, like cost down achievement, the mechanism of randomly matching between hot and cold nodes etc. can be mentioned in the abstract.

2.The study established Nodes-based non-structural model which equipped with fast calculation and good performance in HENS efficiency can be an optimised reference to readers.

Author Response

Dear editors and reviewers,

       Thank you for your letter and the reviewers’ comments concerning our manuscript entitled “A nodes-based non-structural model considering a series structure for heat exchanger network synthesis” (ID processes-828674). All the reviewers are patient and professional, the given comments are valuable for improving our manuscript. We have studied these comments carefully and have made some corrections, which we hope that it could meet with approval. The responses to the reviewers’ comments are written in blue as following. The revised parts are wrote in red in revised manuscript.

1.The abstract is short and clear, it would be better if some main result of this research, like cost down achievement, the mechanism of randomly matching between hot and cold nodes etc. can be mentioned in the abstract.

Response: Thank you for your advice, some quantified data in abstract can be more attractive and more clear for readers to understand the highlight of this manuscript. According to the reviewer’s comment, we added some quantified data about results of cases in this manuscript in Abstract. The revised version is shown in revised manuscript.

2.The study established Nodes-based non-structural model which equipped with fast calculation and good performance in HENS efficiency can be an optimised reference to readers.

Response: Thank you for your comments. We hope this manuscript could offer much idea in heat exchanger network synthesis.

Reviewer 3 Report

The reviewed paper presented the developed node-based non-structural model with a series structure dedicated to heat exchanger network synthesis. The carried out research resulting in the above-mentioned model development is interesting and novel. The title is self-descriptive and represents the content of the manuscript. The abstract provides a clear view of the content of the paper. The authors properly presented the scientific and technological background of the investigated issues in the introduction section. The model itself as well as the research methodology is presented in a sufficient level of detail. The model is validated using the literature data concerning four different cases. The validation cases were properly selected to assure that the developed model will suit systems of different scale. The conclusions are supported by the obtained results.

In my opinion, the authors should better emphasize the novelty of the developed model.

Moreover, I suggest elaborating more on the statement in line 89: “It fits better real-life cases”.

Line 136: should be “formulas” instead of “formulae”.

The units for annual cost presented in the tables are “$/kW/a” and in the text they are “$/a” – it should be unified.

Line 201: I believe it should be “77,958” instead of “910”.

Author Response

Dear editors and reviewers,

       Thank you for your letter and the reviewers’ comments concerning our manuscript entitled “A nodes-based non-structural model considering a series structure for heat exchanger network synthesis” (ID processes-828674). All the reviewers are patient and professional, the given comments are valuable for improving our manuscript. We have studied these comments carefully and have made some corrections, which we hope that it could meet with approval. The responses to the reviewers’ comments are written in blue as following. The revised parts are wrote in red in revised manuscript.

The reviewed paper presented the developed node-based non-structural model with a series structure dedicated to heat exchanger network synthesis. The carried out research resulting in the above-mentioned model development is interesting and novel. The title is self-descriptive and represents the content of the manuscript. The abstract provides a clear view of the content of the paper. The authors properly presented the scientific and technological background of the investigated issues in the introduction section. The model itself as well as the research methodology is presented in a sufficient level of detail. The model is validated using the literature data concerning four different cases. The validation cases were properly selected to assure that the developed model will suit systems of different scale. The conclusions are supported by the obtained results.

1) In my opinion, the authors should better emphasize the novelty of the developed model.

Moreover, I suggest elaborating more on the statement in line 89: “It fits better real-life cases”.

Response: Thank you for your advice. We rewrote this part and added some description about this part as below.

Firstly, NNM model has more flexible setting pattern, series structure is allowed in NNM model, besides, the number of series nodes can be freely set without any limitation. Secondly, the stage concept is discarded, and the positions of heat exchangers are quantified by the positions of nodes, thus, more flexible matching pattern than other model, so that it could offer many potential structures for later optimization. Thirdly, only variables concerning nodes are used for positions’ qualifying, which improves the computational efficiency. Fourthly, in real life engineering, the series or parallel heat exchanger structures always exist in a system. NNM model could search the best solution in a shorter computational time, which is benefit for saving cost and resources, so it fits better real-life cases.

2) Line 136: should be “formulas” instead of “formulae”.

Response: According to the reviewer’s comment, we rechecked some dictionary, the “formulae” and “formulas” are both plural forms of “formula”. The “formulas” may more suitable for research paper, therefore, we substituted the “formulae” to “formulas” in whole manuscript.

3) The units for annual cost presented in the tables are “$/kW/a” and in the text they are “$/a” – it should be unified.

Response: Thank you for your comment, “$/kW/a” is the unit of annual cost of hot or cold utilities, it is the cost of per kW per year. “$/a” is the unit of total annual cost. They represents different meaning, so it cannot unified.

4) Line 201: I believe it should be “77,958” instead of “910”.

Response: The original sentence want to express the comparison of costs obtained by different models, “77,958” is the cost obtained by NNM model proposed in this manuscript while “910” is the difference costs between Figure 3 and Figure 4. Therefore, we revised this sentence as below, we hope the revised version can be clear for readers:

The TAC in Figure 4 is 77,048 $/a, which is lower 910 $/a than that in Figure 3, it was achieved in 521.92 s.

Round 2

Author Response

Response: Thank you for your comments, according to the comment of reviewer, we changed the method of introducing the formulas in this manuscript. We hope the revised version could meet the requirement. We rechecked the whole manuscript, we found that some abbreviations are wrongly written in the bracket. We have corrected this kind of mistake.

Round 3

Reviewer 1 Report

Authors made suggested correction and improve the paper.